# *Trans* Isomeric Fatty Acids in Children and Young Adults with Type 1 Diabetes Mellitus

**DOI:** 10.3390/nu17111907

**Published:** 2025-06-01

**Authors:** Éva Szabó, Tamás Marosvölgyi, Krisztina Mihályi, Szimonetta Lohner, Tamás Decsi

**Affiliations:** 1Department of Pediatrics, Clinical Centre, University of Pécs, 7623 Pécs, Hungary; marosvolgyi.tamas@pte.hu (T.M.); mihalyi.krisztina@pte.hu (K.M.); lohner.szimonetta@pte.hu (S.L.); 2Department of Biochemistry and Medical Chemistry, Medical School, University of Pécs, 7624 Pécs, Hungary; 3Institute of Bioanalysis, Medical School, University of Pécs, 7624 Pécs, Hungary; 4Department of Emergency Medicine, Medical School, University of Pécs, 7624 Pécs, Hungary; 5Department of Public Health Medicine, Medical School, University of Pécs, 7624 Pécs, Hungary

**Keywords:** *trans* fatty acid, type 1 diabetes mellitus, children, young adults, diabetic ketoacidosis

## Abstract

Background/Objectives: Dietary guidelines recommend limiting *trans* fatty acid (TFA) intake to avoid adverse health effects. However, the impact of TFA intake in type 1 diabetes mellitus (T1DM) remains unclear. The aim of the present study was to investigate the levels of TFAs in plasma and erythrocyte membrane lipids of young diabetic patients and healthy controls. Methods: Data were re-analyzed from three case-control studies including diabetic children (*n* = 40, mean age: 12.0 years), diabetic young adults (*n* = 34, mean age: 21.8 years), and children with diabetic ketoacidosis (DKA, *n* = 9, mean age: 16.0 years). In these studies, TFA data were quantified by gas chromatography, but data have not yet been published. Results: Diabetic young adults and diabetic children had significantly lower TFAs in plasma lipids compared to healthy controls (sum of TFA in plasma sterol esters: 0.54 [0.34] versus 0.64 [0.37] and 0.51 [0.13] versus 0.65 [0.29], %, median [interquartile range], *p* < 0.05). However, children with DKA had significantly higher TFA levels in almost all plasma lipid fractions than the other two diabetic groups. Several negative correlations were observed between TFA and *n*-3 and *n*-6 long-chain polyunsaturated fatty acid levels in all groups, especially in the erythrocyte membrane lipid fractions. However, in the plasma fractions the correlation was less clear; both positive and negative correlations were found in each of the groups studied. Conclusions: Lower TFA values in young adults and children with diabetes may be associated with dietary patterns lower in TFAs, while elevated TFA values in DKA may be linked to challenges in adherence to dietary guidelines.

## 1. Introduction

The availability and metabolism of fatty acids, as well as their health implications, have already been the subject of extensive investigations in type 1 diabetes mellitus (T1DM). In recent years, the role of *trans* isomeric fatty acids (TFAs) in T1DM has emerged as a topic of growing interest; however, the available literature on this topic remains relatively limited and controversial [1,2,3,4].

TFAs are unsaturated fatty acids with at least one unsaturated, non-conjugated double bond in the *trans* configuration. There are two main sources of TFAs in the diet: a) industrially produced TFAs (iTFAs), which primarily originate from commercially hydrogenated vegetable and marine oils, and b) natural, ruminant-origin TFAs (rTFAs), including dairy and meat fats that contain TFAs from ruminant biohydrogenation [5]. The most prominent iTFAs are C18:1n-9*t* (elaidic acid) and C18:2n-6*tt* (linoelaidic acid), while C18:1n-7*t* (vaccenic acid) is the most prominent rTFA. It should be noted that the same TFA isomers can be present in ruminant and industrial fats, but their proportions may differ. Margarine, baking fats, and fats used for frying may contain up to 40–50% TFAs, while ruminant fats typically contain 1–8% [6]. In consideration of the detrimental impact of iTFAs on human health, the European Food Safety Authority recommends that TFA intake should be limited to the lowest feasible level in a nutritionally balanced diet [7]. In accordance with this guideline, the World Health Organization (WHO) advises that, in order to prevent chronic diseases, consumption of iTFA should be restricted to less than 1% of daily energy intake [8]. In recent decades, regulations have been implemented on a global scale to reduce TFA intakes from industrial sources [9]. Despite these regulations, high levels of TFAs can still be detected in certain food groups in many countries [10].

Currently available evidence indicates that the consumption of iTFAs has a detrimental impact on multiple cardiovascular risk factors and significantly elevates the risk of cardiovascular disease (CVD) and mortality [11,12]. This is at least partially due to their capacity to induce vascular inflammation and impair insulin-mediated nitric oxide production in endothelial cells [13]. iTFAs have been demonstrated to increase total cholesterol and low-density lipoprotein cholesterol (LDL-C) and decrease high-density lipoprotein cholesterol (HDL-C) in the blood, leading to a more unfavorable lipid profile and playing a key role in the development of cardiovascular diseases [14]. In T1DM patients, early-onset atherosclerosis manifests more often even among subjects in the younger age group compared to healthy controls. In its development, iTFAs may play a role, but research on this topic is rather limited.

In contrast, the consumption of rTFAs may not exert any adverse effects [14,15], or may even be beneficial [16,17]. However, practically no studies investigated the effect of limiting TFA intake on the risk factors of CVD among healthy children and young adults [18].

Another potential untoward effect of TFAs might be their competition with essential fatty acids for the enzymes involved in the conversion of essential fatty acids to their longer-chain metabolites. Specifically, a negative correlation was observed between TFAs and long-chain polyunsaturated fatty acids (LCPUFAs) in the perinatal period [19,20].

Evidence-based dietary management of patients with T1DM involves the adjustment of insulin dosage to carbohydrate intakes [21,22]. In these patients, it is highly important to reduce the risk of cardiovascular diseases by following a healthy dietary pattern, in particular the Mediterranean or DASH (Dietary Approaches to Stop Hypertension) diets. Regarding lipid intake, it is essential to avoid high fat intakes and to replace saturated fats with unsaturated fats while avoiding iTFA intakes [22].

It is a common belief that children and young adults with T1DM consume healthier diets than their non-diabetic counterparts. However, this may not always be the case, as dietary restrictions and the strict diabetic dietary recommendations may not necessarily result in optimal nutritional intakes. Although Australian children with T1DM consumed a greater quantity of vegetables than healthy controls, both groups exceeded the national recommendation for saturated and TFA intake [23]. A review also indicated that children with T1DM tend to consume higher amounts of saturated fatty acids and adhere to less healthy dietary patterns [24].

Given the potential adverse effects of TFAs on human health, our present study aimed to examine, using a cross-sectional method, whether there is a difference in the blood levels of TFAs between children and young adults (1–32 years) with T1DM and healthy controls. A further objective was to ascertain whether elevated TFA values disrupt the metabolism of *n*-3 and *n*-6 fatty acids, leading to inverse correlations with *n*-3 and *n*-6 LCPUFA values. Additionally, the investigation aimed to explore the association between TFA levels and blood lipids in a group of diabetic patients and their healthy counterparts, all of them without already existing cardiovascular complications.

## 2. Materials and Methods

Our research group has previously published three papers describing fatty acid composition of plasma and erythrocyte membrane lipids in children with T1DM [25], in young adults with T1DM [26], and in children with T1DM during and after diabetic ketoacidosis (DKA) [27]. At the time of these publications, our research activity was focused on LCPUFAs. Although the TFAs were also quantified in all these studies, they were not included in the papers published [25,26,27].

Eligibility criteria for participants, the settings and locations of data collection, and the methods of sample collection and management have been described in detail in our previous publications [25,26,27]. However, the most important steps are also discussed here. The group of children with T1DM comprised diabetic children aged between 1 and 18 years who presented to the endocrinology department for a control examination between July and December 2000. The children selected for this study did not suffer from any other comorbidities besides T1DM (e.g., celiac disease, thyroid disease), and all of them were on insulin therapy. No additional exclusion criteria were implemented. The control group was matched for age and sex and was free of chronic disease. In the T1DM young adult group, young adults aged 18–32 years who did not have any comorbidities other than T1DM were recruited. The controls were matched for age and sex and were free of chronic disease. The participants in this study were recruited between July and December of 2000. Blood samples were collected from the antecubital vein in the morning. The plasma was removed and frozen within 30 min, and the erythrocyte mass was washed with isotonic sodium chloride solution three times. The final sediment was dissolved in 1 mL of distilled water at ambient temperature for 20 min. Subsequently, 2 mL of ice-cold isopropyl alcohol, containing 0.5% butylated hydroxytoluene (BHT) as an antioxidant, was added drop-wise during continuous shaking. The samples were stored in a −80 °C freezer until further analysis.

Triacylglycerol and cholesterol were measured using an enzymatic kit (Boehringer Mannheim, Mannheim, Germany). Glycated hemoglobin was measured using a DiaSTAT HPLC device (Bio-Rad Laboratories Diagnostics Group, Hercules, CA, USA). All samples were thawed only once to prevent fatty acids from degradation. After lipid extraction the lipid classes from plasma and erythrocyte membrane samples were separated on thin layer chromatography. Fatty acid methylesters were analyzed by high-resolution capillary gas–liquid chromatography using a Finnigan 9001 gas chromatograph (Finnigan/Tremetrics Inc., Austin, TX, USA) with split injection (ratio: 1 to 25), automatic sampler (A200SE, CTC Analytic, Switzerland), and flame ionization detector with a DB-23 cyanopropyl column of 40 m length (J&W Scientific, Folsom, CA, USA). Peak identification was verified by comparison with authentic standards. Fatty acid results were expressed as the percentage (by weight) of fatty acids detected with a chain length between 12 and 24 carbon atoms. Three *trans* isomers were determined from the plasma and erythrocyte membrane lipid fractions: *trans* hexadecenoic acid (C16:1n-7*t*), *trans* octadecenoic acid (C18:1n-7/9*t*), and *trans* octadecadienoic acid (C18:2n-6*tt*). The sum of *trans* isomeric fatty acids was calculated by adding these three identified *trans* isomers.

The statistical analysis was conducted using IBM SPSS Statistics 28.0 software (IBM Corporation, Armonk, NY, USA). Mann–Whitney’s two-sided rank test was used to compare the fatty acids between diabetic patients and healthy controls, while the Kruskal–Wallis test with Bonferroni correction was employed to determine statistical significance between children during DKA, diabetic children, and their healthy controls. Spearman’s ρ correlation coefficients were used to determine the correlations between the concentrations of each TFA (C16:1n-7*t*, C18:1n-7/9*t*, and C18:2n-6*tt*), as well as their summarized values, and the main *n*-3 and *n*-6 PUFAs and blood triacylglycerol and cholesterol values. The statistical significance of the results was set at *p* < 0.05.

## 3. Results

### 3.1. Trans Fatty Acids in Diabetic Young Adults

The basic anthropometric and clinical data of the investigated diabetic groups and their controls are described in Appendix A. *Trans* fatty acid values in young adults with T1DM (*n* = 34; age: 21.8 [3.1] years, mean [SD]; HgbA1c: 8.86% [2.67]; 19 males and 15 females) and healthy controls (*n* = 36; age: 22.6 [2.6] years; 19 males, 17 females) are described in Table 1. Although all TFA values in all lipid fractions were low in both groups, diabetic young adults exhibited even lower levels. Levels of C16:1n7*t* were significantly lower in the plasma phospholipid (PL) and triacylglycerol (TG) as well as in erythrocyte phosphatidylethanolamine (PE) fractions in diabetic adults compared to healthy controls. Similarly, the sum of all TFAs was significantly lower in diabetic adults than in the control group, both in the plasma sterol ester (STE) and non-esterified fatty acid (NEFA) fractions. In contrast, no significant differences in the C18:1n7/9*t* and C18:2n-6*tt* values were observed between diabetic adults and healthy controls in either fractions.

### 3.2. Trans Fatty Acids in Diabetic Children

Children with T1DM (*n* = 40; age: 12.03 [3.90] years, mean [SD]; HgbA1c: 9.24% [2.29]; 12 boys and 28 girls) and healthy controls (*n* = 40; age: 12.45 [3.49] years, mean [SD]; 18 boys and 22 girls) both exhibited relatively low values of TFAs in both plasma and erythrocyte membrane lipid fractions (Table 2). Values of C16:1n-7*t* were significantly lower in plasma TG and NEFA fractions of diabetic children than in controls. Similarly, values of C18:1n-7/9*t* and the sum of total TFAs in the plasma STE fraction were significantly lower in diabetic children than in healthy controls. However, the differences in TFA levels between the two groups were less pronounced than the corresponding differences seen in young adults. In contrast, values of C18:1n-7/9*t* and C18:2n-6*tt* in the plasma NEFA fraction, as well as the values of C18:1n-7/9*t* and C18:2n-6*tt* and the sum of total TFAs in the erythrocyte membrane PE fraction, were significantly higher in diabetic children than in healthy controls. However, no significant differences were observed in either plasma PL or erythrocyte phosphatidylcholine (PC) fraction between diabetic children and controls in either *trans* fatty acids.

### 3.3. Trans Fatty Acids in Diabetic Children with and Without Diabetic Ketoacidosis

During DKA in children with T1DM (*n* = 9; age: 16.71 [4.27] years; HgbA1c: 13.4% [2.8]; 4 boys and 5 girls), higher TFA values were measured in the plasma lipid fractions than in our two previous studies. With the exception of C16:1n-7*t* values in the PL and C18:2n-6*tt* in the NEFA fraction, the other *trans* isomers in the DKA group were significantly different from those in the diabetic children and controls. Levels of C16:1n-7*t* levels were significantly lower in children with DKA in comparison to diabetic children and healthy controls in the TG and STE fractions (Figure 1). In contrast, the C18:1n-7/9*t*, C18:2n-6*tt*, and sum of total TFAs were significantly higher in the PL, TG, and STE fractions during DKA than in diabetic children without DKA and healthy controls. In the NEFA fraction, C18:1n-7/9*t* and the sum of total TFAs were significantly higher in the DKA group than in the other two groups (Figure 1).

### 3.4. Correlation Between Trans Isomeric and Polyunsaturated Fatty Acids

In young adults with T1DM as well as in their healthy controls, several negative correlations were found between the most important *n*-6 and *n*-3 fatty acid values and TFAs (Appendix A). In young adults with T1DM, significant inverse correlations were found between the most important *n*-6 metabolite, arachidonic acid (C20:4n-6, AA), and C18:1n-7/9*t* and the sum of total TFAs in the erythrocyte PE fraction. Similarly, the values of the most important *n*-3 LCPUFA, docosahexaenoic acid (C22:6n-3, DHA), correlated inversely to C18:1n-7/9*t* values in the PE fraction. In the plasma TG fraction, there were strong negative correlations between docosapentaenoic acid (C22:5n-3, DPA) values and all investigated TFA values as well as the sum of total TFAs.

In the controls, significant inverse associations were found between, on the one hand, AA values and, on the other hand, C16:1n-7*t*, C18:1n-7/9*t*, and the sum of total TFAs in the erythrocyte PC fraction (Appendix A). Similarly, the values of the metabolite of AA, C22:5n-6, also correlated significantly and inversely to C16:1n-7*t*, C18:1n-7/9*t*, and the sum of total TFA values in the erythrocyte PC fraction. With regard to *n*-3 fatty acids, in the control group, only C16:1n-7*t* values showed an inverse relationship with the levels of the major *n*-3 metabolite, DHA, as well as *n*-3 PUFAs and *n*-3 LCPUFAs in the erythrocyte PC fraction (Appendix A). However, in addition to the negative correlations, we also saw many significant positive correlations between TFAs and cis-isomeric PUFA values in both young adults with T1DM and their healthy controls (Appendix A).

In children with T1DM and their healthy controls, we saw slightly more consistent correlations, especially for the plasma STE fraction (Appendix A). Values of the essential *n*-6 fatty acid, linoleic acid (C18:2n-6, LA), and the *n*-6 PUFA correlated significantly and inversely to C16:1n-7*t*, C18:2n-6*tt*, and the sum of total TFAs in diabetic children and with C16:1n-7*t*, C18:1n-7/9*t*, and the sum of total TFAs in healthy controls in the STE fraction. Although no significant correlation was found between AA and TFA values in diabetic children in the plasma lipid fractions, the values of the major *n*-6 metabolite, AA, in healthy controls were significantly and inversely correlated with C18:1n-7/9*t*, C18:2n-6*tt*, and the sum of total TFAs in both plasma TG and STE fractions. Except for DPA, the other *n*-3 fatty acid levels were significantly and inversely related to C18:1n-7/9*t* in the STE fraction in control children (Appendix A). However, similar to the findings in young adults, we also saw several significant positive correlations between cis and *trans* isomers in children (Appendix A).

The most ambiguous correlations between cis and *trans* isomer values were observed in diabetic children during and after DKA (Appendix A). In children during DKA, significant inverse correlations were observed between the levels of LA, C22:4n-6, *n*-6 PUFA, and *n*-6 LCPUFA and both C18:1n-7/9*t* and the sum of total TFAs in the plasma TG fraction. Moreover, significant negative correlations were observed between AA and *n*-6 LCPUFA values and both C18:2n-6*tt* and the sum of total TFAs levels in the plasma STE fraction. After DKA, there were inverse correlations between AA and *n*-6 PUFA values and C18:1n-7/9*t* and the sum of total TFAs in the plasma PL fraction. In the plasma TG fraction, C18:3n-3 (alpha-linolenic acid, ALA) values correlated inversely to C18:1n-7/9*t*, C18:2n-6*tt*, and the sum of total TFA values during DKA. After DKA, a significant negative association was observed between *n*-3 LCPUFA and C18:2n-6*tt* and the sum of total TFA values in the plasma TG fraction (Appendix A).

### 3.5. Correlation Between Trans Isomeric Fatty Acids and Blood Lipids

In young adults with T1DM, only one significant inverse correlation was found between blood triacylglycerol concentrations and C16:1n-7*t* in the PL fraction (r = −0.381, *p* < 0.05). In contrast, in the control group significant negative correlations were found between blood cholesterol concentrations and C18:1n-7/9*t* in the STE and NEFA fraction (r = −0.371, *p* < 0.05 and r = −0.550, *p* < 0.01, respectively). Triacylglycerol values also showed significant inverse correlation with C16:1n-7*t* in the TG (r = −0.339, *p* < 0.05) and C18:1n-7/9*t*, as well as the sum of TFAs in the PE fraction (r = −0.353, *p* < 0.05 and r = −0.331, *p* < 0.05, respectively).

In children with T1DM, significant positive correlation was found between cholesterol concentrations and C18:2n-6*tt* values in the TG fraction (r = 0.364, *p* < 0.05), while triacylglycerol values correlated significantly and inversely to C16:1n-7*t* in the TG and PC fractions (r = −0.527, *p* < 0.001 and r = −0.399, *p* < 0.05, respectively). In the control children, blood cholesterol values correlated significantly and positively to C18:1n-7/9*t* and the sum of total TFA values in the PL (r = 0.353, *p* < 0.05 and r = 0.325, *p* < 0.05), NEFA (r = 0.344, *p* < 0.05 and r = 0.340, *p* < 0.05), and PC fractions (r = 0.388, *p* < 0.05 and r = 0.356, *p* < 0.05), but no significant correlations were detected between blood triacylglycerol concentrations and TFA values either in plasma or in erythrocyte membrane lipid fractions.

## 4. Discussion

The present study somewhat surprisingly showed that certain plasma TFA values were significantly lower in diabetic young adults and children than in their healthy controls. Conversely, in children with DKA, TFA values were significantly elevated. These results suggest that T1DM patients with more stable blood sugar levels may have generally lower TFA intake compared to their healthy peers. However, it is important to emphasize that both groups had very low levels of total TFA in all lipid fractions, so the statistically significant difference may have little physiological significance. But the present cross-sectional study design is inadequate for answering this question, and longer follow-up studies would be required to address it. The usual dietary restrictions in diabetes include reduced intake of fast food, TFA-rich cookies, chips, and baked goods, which may well result in lower dietary intakes of TFAs. However, a study of Italian adolescents revealed that only 6.2% of the participants met all three health-related recommendations (specifically: high adherence to Mediterranean diet, high physical activity, and low sedentary behavior). Meanwhile, 68.2% of the participants exhibited at least two unhealthy lifestyle habits, with no observed gender difference. As the number of unhealthy habits increased, there was a significant decrease in the quality of life total score and its sub-scales scores (including diabetes symptoms) [28]. A Polish research group showed that in young adults with T1DM there may be a gender difference in the intake of different food groups. The male participants exhibited lower Healthy Diet Index scores and a significantly higher frequency of consumption of alcohol, sweets (including cakes, chocolate bars, and other confectionery) and fast food (including French fries, hamburgers); all these differences may have led to elevated TFA intakes [29].

This observation was further corroborated by a subsequent review by Patton [30], who systematically collected studies reporting on usual dietary intake and/or adherence to diet therapy in young patients with T1DM (age: 0 to 22 years). Based on the 23 relevant papers included in this review, rates of adherence to recommended eating behaviors ranged from 21% to 95%, indicating thereby that a significant proportion of young individuals with T1DM encounter challenges in adhering to dietary guidelines for their condition. A subsequent study on American adolescents with T1DM revealed that their dietary fat intakes exceeded the recommended intakes in dietary guidelines and was significantly higher compared to a matched group of healthy controls. Additionally, TFA intakes of diabetic subjects were found to be significantly higher than those of healthy controls; however, the influence of gender was non-significant [31]. In contrast, Australian diabetic children consumed a diet quite similar to their matched healthy controls without any significant difference in macro- and micronutrient intakes; however, neither group met the national dietary guidelines. The overconsumption of saturated/*trans* fatty acids (and sodium) was observed in both groups, indicating an unhealthy dietary pattern [23].

Vitale et al. [2] investigated the dietary intake of TFAs and the fatty acid composition of serum PLs in young Italian patients with T1DM aged 18 to 30 years and healthy, age-matched controls. The dietary records indicated that the intake of TFAs was significantly lower in the T1DM patient group compared to the control group. However, the actual intake levels were rather low in both groups (0.13 [0.13] vs. 0.22 [0.14] % of calories, mean [SD]). Subsequent analysis revealed that there were no statistically significant differences between the two groups in the TFA content of serum PLs. Consistent with this study, we did not find any statistically significant differences in the investigated TFA levels between T1DM patients and controls in the plasma PL fraction, with one exception (C16:1n-7*t* in young adults). However, in our study, we measured lower TFA levels, approximately half of those reported in the Italian study, which suggests the possibility of a lower dietary intake in Hungarian children and young adults compared to other populations. In contrast, the DKA group showed higher TFA levels compared to the previous two diabetic groups, suggesting the potential for an unhealthier diet in these children. TFAs are exclusively derived from dietary sources, indicating that these children were likely to have a higher consumption of foods with higher TFA values. These findings indicate that in comparison with diabetic children and young adults, children with DKA exhibit less stable carbohydrate metabolism (as indicated, e.g., by HbA1c values). One potential explanation for this finding is the difference in dietary intakes of these patients and the potential lower adherence to dietary and lifestyle recommendations. Although TFAs are derived from diet, several confounding factors may also affect blood fatty acid levels. These include physical activity, patient compliance, type and amount of insulin treatment, stage of puberty, and length of illness. In the present study, data for certain confounders (e.g., physical activity, type of insulin treatment) were not available. For other confounders (e.g., stage of puberty, length of disease), the number of cases was insufficient to conduct subgroup analyses. Therefore, further studies with a larger number of participants are required to determine whether these confounding effects have a significant impact on blood TFA levels and whether there is a difference between T1DM subjects who demonstrate adequate compliance and diabetes control compared to healthy controls.

TFA levels may be influenced by factors beyond diet, such as the time since diagnosis, i.e., the duration of diabetes. For instance, children diagnosed with T1DM less than one year before had significantly higher plasma NEFA C18:1n-9*t* levels than children diagnosed more than one year before and healthy controls [3]. The authors of this study excluded differences in dietary intakes of TFAs as a potential cause, but raised the possibility of an altered microbiome due to the development of diabetes as a trigger. This finding aligns with the results of our recent study [32], in which changes in the gut microbiome significantly affected fecal fatty acid composition, including TFAs, in a rat model. A recent cohort study found altered microbiome as well as serum and fecal metabolome in adult-onset T1DM patients compared to T2DM and healthy controls [33]. Therefore, it can be hypothesized that host–microbiota–metabolite interactions might be involved in the pathogenesis of T1DM. However, this hypothesis requires corroboration by large-scale studies.

Several previous studies in adult patients have investigated the possible effects of higher TFA intakes on blood glucose, insulin sensitivity, and blood lipid levels. In an in vitro cell study [34], both cis and *trans* C16:1n-7 stimulated insulin secretion through the G protein-coupled receptor, which may be primarily involved in the pathogenesis of type 2 diabetes mellitus (T2DM). However, an interesting finding was that C16:1n-7*t* reduced cell survival much more than the cis isomer, suggesting a direct beta-cell toxicity of high TFA intake. However, this hypothesis needs to be confirmed by further animal and human studies. According to a meta-analysis by Aronis et al. based on the results of seven placebo-controlled randomized clinical trials, higher TFA intakes were associated with significantly increased LDL-C and total cholesterol levels, as well as with decreased HDL-C levels [35]. In another meta-analysis summarizing the results of observational studies, higher intakes of total and industrial TFAs were significantly associated with the relative risk of all-cause mortality, coronary heart disease (CHD), and CHD mortality [11]. At the same time, higher rTFA intakes were related to significant reduction in the relative risk of T2DM, based on the pooled results of five studies. In addition, a recent meta-analysis found with moderate certainty an increased risk of coronary events in a general population with the highest TFA intake compared to the lowest [36]. In the present study, some positive correlations were found between TFA values and some blood lipids in diabetic patients and healthy controls, but the associations were weak and partly controversial.

Given the adverse human health effects of iTFAs, the WHO recommends limiting iTFA consumption to less than 1% of daily energy intake to prevent chronic diseases [37]. Previous studies have shown that global intakes of TFAs decreased significantly over the past two decades [9], as a result of both voluntary and regulatory measures to reduce iTFA content in foods. According to the most recent study, TFA intakes were below 0.8% in most regions of the world, with the highest reported intakes in North, Central, and Latin America, ranging from 0.8% to 1.2% [38]. A decrease in dietary iTFA intake has been shown to result in a decrease in total TFA content in blood samples, as demonstrated in a cross-sectional study. The regulation of food iTFA content in the USA led to a substantial decrease in plasma PL TFA content among patients with T2DM over time, between September 2002 and May 2004. However, it is important to note that the overall impact of the potential beneficial effect of TFA reduction on reduced population-wide cardiovascular risk is influenced also by the composition of the fatty acid groups that replace TFAs [39].

A number of previous studies described the adverse effects of TFAs on the metabolism of *n*-3 and *n*-6 PUFAs during the perinatal period [19,20,40]. However, there is a paucity of information regarding whether this significant interference occurs at a later stage in life or in specific diseases. To the best of our knowledge, there is only one study that investigated the relationship between cis and *trans* isomers in the erythrocyte membrane lipids in patients with T1DM. The findings revealed a significant inverse correlation between all C18:1*t* isomers and *n*-6 LA, AA, while positive relationships were observed with *n*-3 DPA and DHA. In contrast, C16:1n-7*t* exhibited an inverse correlation with LA and DHA values. The authors also demonstrated a direct and significant relationship between erythrocyte C18:1*t* content and the presence of three or more carotid plaques (OR 1.51 [1.05–2.16], *p* = 0.026; 0.1% increase of total FAs), which serves as a predictor of future cardiovascular events [4]. This observation underscores the significance of maintaining low dietary TFA intake in children and young adults with T1DM. The findings of the present study indicate that the adverse impact of TFAs on the metabolism of physiologically relevant *n*-3 and *n*-6 PUFAs is less pronounced in young adults and children with diabetes, as well as in their matched controls, when compared to the findings of earlier studies conducted during the perinatal period. Nevertheless, a greater number of negative correlations were identified between TFAs and *n*-6 fatty acids as compared to *n*-3 fatty acids. It is noteworthy that the levels of TFAs during and after DKA were higher compared to the levels observed in the other two diabetic groups and their matched controls.

The present study has several strengths and limitations. One limitation is that the participants did not complete a dietary questionnaire, so we do not have accurate information on intake of fatty acids and other macronutrients. Consequently, we can only hypothesize a lower intake of TFAs in the group of Hungarian children and young adults with T1DM, as the human body does not synthesize these fatty acids; thus, the measured fatty acid values directly reflect dietary intakes. Moreover, though we do not have the exact TFA content of the foods available in Hungary at the time of our studies, still we can speculate that TFA values were lower in Hungary than in several international samples measured at similar times. Hungary was among the first countries in Europe to implement regulations limiting the maximum TFA content of foods in 2013, a crucial step in addressing the issue. In the diabetic patients included in the original studies, higher triglyceride, cholesterol, or HbA1c levels were not exclusion criteria and may have been confounding factors in our results. Consequently, further investigation is required to validate our findings using more rigorous inclusion criteria and to exclude potential confounding variables. A further limitation is that the number of children with diabetes and children with DKA differed significantly. Although the comparisons have been statistically corrected for sample size, the results should be treated with some caution, and further studies should confirm the differences seen in TFA values. A strength of the study is that the blood samples were analyzed in the same laboratory, using the same instruments and methodologies. An additional strength of this study is the inclusion of diverse patient cohorts with T1DM, encompassing children and young adults who developed the disease. Moreover, to the best of our knowledge, this is the first study to investigate the association between cis and *trans* isomeric fatty acids in diabetic children and young adults without cardiovascular complications.

## 5. Conclusions

On the basis of the three distinct studies reported here, we conclude that young Hungarian diabetic adults are not at increased risk for higher TFA intakes. Indeed, our data suggest that the diet of these diabetic adults may contain lower levels of some TFAs than the diet of the general population. In young diabetic children the situation is not as clear as in young diabetic adults; however, based on our present data, Hungarian diabetic patients investigated here were relatively well educated in terms of avoiding TFA intake. However, it should be noted that the small sample size and the study design limit the generalizability of our findings to children and adolescents with T1DM in other countries. The findings of this study also indicate that the hypothesized inhibitory effect of TFAs on *n*-3 and *n*-6 PUFA metabolism appears to be less pronounced in diabetic patients than those reported in the literature for young infants. In these cross-sectional studies, the investigation was focused exclusively on the plasma and erythrocyte membrane lipid TFA content in patients with T1DM and their healthy controls. However, further studies are required to ascertain whether the diets of children and adolescents with T1DM exhibit a distinct fatty acid composition compared to their healthy peers, particularly with regard to TFAs. Furthermore, potential influencing factors (duration of diabetes, efficacy of diabetes control, physical activity, Tanner stage, dyslipidemia) need to be investigated to determine whether there are factors other than diet that may significantly influence TFAs in children and adolescents with T1DM.

## Figures and Tables

**Figure 1 nutrients-17-01907-f001:**
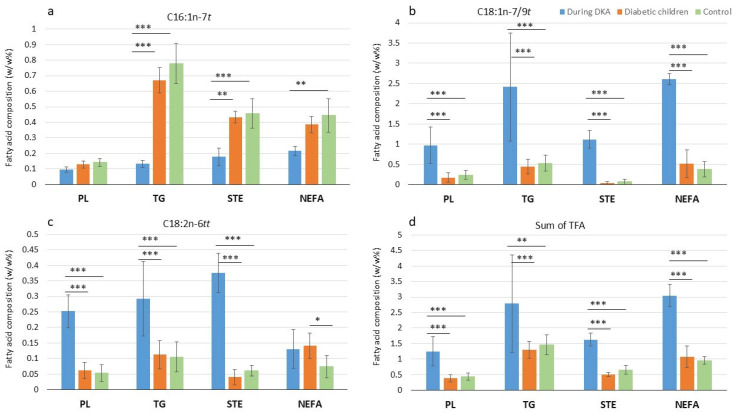
Values of (**a**) C16:1n-7*t*, (**b**) C18:1n-7/9*t*, (**c**) C18:2n-6*tt* and (**d**) sum of *trans* fatty acids in diabetic children with diabetic ketoacidosis (*n* = 9, blue), diabetic children (*n* = 40, orange), and controls (*n* = 40, green) in plasma phospholipids (PL), triacylglycerols (TG), sterol esters (STE), and non-esterified fatty acids (NEFA). Data are in median (IQ); asterisks mean significant difference between two groups: *: *p* < 0.05, **: *p* < 0.01, *** *p* < 0.001.

**Table 1 nutrients-17-01907-t001:** *Trans* fatty acid values in plasma and erythrocyte membrane lipid fractions in diabetic adults and age-matched healthy controls.

Lipid Fraction	*Trans* Fatty Acids	Diabetic Adults (*n* = 34)	Controls (*n* = 36)	*p*
Plasma phospholipid	C16:1n-7*t*	0.12 (0.05)	0.16 (0.08)	**
	C18:1n-7/9*t*	0.19 (0.16)	0.21 (0.25)	
	C18:2n-6*tt*	0.08 (0.04)	0.08 (0.06)	
	Sum of *trans*	0.43 (0.17)	0.46 (0.33)	
Plasma triacylglycerol	C16:1n-7*t*	0.69 (0.19)	0.79 (0.16)	*
	C18:1n-7/9*t*	0.52 (0.36)	0.42 (0.48)	
	C18:2n-6*tt*	0.12 (0.12)	0.10 (0.06)	
	Sum of *trans*	1.41 (0.61)	1.37 (0.49)	
Plasma sterol ester	C16:1n-7*t*	0.42 (0.22)	0.49 (0.23)	
	C18:1n-7/9*t*	0.06 (0.07)	0.08 (0.16)	
	C18:2n-6*tt*	0.06 (0.04)	0.08 (0.08)	
	Sum of *trans*	0.54 (0.34)	0.64 (0.37)	*
Plasma non-esterified fatty acid	C16:1n-7*t*	0.56 (0.30)	0.50 (0.44)	
	C18:1n-7/9*t*	0.27 (0.51)	0.56 (0.59)	
	C18:2n-6*tt*	0.08 (0.09)	0.08 (0.05)	
	Sum of *trans*	1.14 (0.62)	1.36 (0.46)	*
Erythrocyte membrane phosphatidylcholine	C16:1n-7*t*	0.12 (0.04)	0.12 (0.04)	
	C18:1n-7/9*t*	0.20 (0.17)	0.18 (0.14)	
	C18:2n-6*tt*	0.08 (0.02)	0.07 (0.03)	
	Sum of *trans*	0.42 (0.17)	0.37 (0.16)	
Erythrocyte membrane phosphatidylethanolamine	C16:1n-7*t*	0.11 (0.04)	0.13 (0.06)	**
	C18:1n-7/9*t*	0.20 (0.12)	0.17 (0.19)	
	C18:2n-6*tt*	0.04 (0.02)	0.04 (0.01)	
	Sum of *trans*	0.36 (0.17)	0.38 (0.22)	

Values are in median (IQR). Asterisks indicate significant differences (*: *p* < 0.05; **: *p* < 0.01) between diabetic adults and healthy controls.

**Table 2 nutrients-17-01907-t002:** *Trans* fatty acid values in plasma and erythrocyte membrane lipid fractions in diabetic children and in age-matched healthy controls.

Lipid Fraction	*Trans* Fatty Acids	Diabetic Children (*n* = 40)	Controls (*n* = 40)	*p*
Plasma phospholipid	C16:1n-7*t*	0.13 (0.05)	0.14 (0.05)	
	C18:1n-7/9*t*	0.18 (0.22)	0.24 (0.21)	
	C18:2n-6*tt*	0.06 (0.05)	0.05 (0.05)	
	Sum of *trans*	0.39 (0.23)	0.44 (0.24)	
Plasma triacylglycerol	C16:1n-7*t*	0.67 (0.16)	0.78 (0.26)	*
	C18:1n-7/9*t*	0.45 (0.36)	0.53 (0.39)	
	C18:2n-6*tt*	0.11 (0.09)	0.11 (0.10)	
	Sum of *trans*	1.30 (0.56)	1.46 (0.64)	
Plasma sterol ester	C16:1n-7*t*	0.43 (0.08)	0.46 (0.19)	
	C18:1n-7/9*t*	0.04 (0.08)	0.07 (0.13)	*
	C18:2n-6*tt*	0.04 (0.05)	0.06 (0.03)	
	Sum of *trans*	0.51 (0.13)	0.65 (0.29)	*
Plasma non-esterified fatty acid	C16:1n-7*t*	0.39 (0.11)	0.44 (0.22)	*
	C18:1n-7/9*t*	0.52 (0.68)	0.39 (0.38)	*
	C18:2n-6*tt*	0.14 (0.08)	0.08 (0.07)	**
	Sum of *trans*	1.08 (0.70)	0.95 (0.24)	
Erythrocyte membrane phosphatidylcholine	C16:1n-7*t*	0.12 (0.05)	0.12 (0.04)	
	C18:1n-7/9*t*	0.21 (0.17)	0.18 (0.14)	
	C18:2n-6*tt*	0.08 (0.05)	0.07 (0.03)	
	Sum of *trans*	0.41 (0.21)	0.37 (0.16)	
Erythrocyte membrane phosphatidylethanolamine	C16:1n-7*t*	0.13 (0.10)	0.14 (0.11)	
	C18:1n-7/9*t*	0.36 (0.26)	0.23 (0.14)	*
	C18:2n-6*tt*	0.04 (0.02)	0.03 (0.02)	*
	Sum of *trans*	0.52 (0.29)	0.43 (0.23)	*

Values are in median (IQR). Asterisks indicate significant differences (*: *p* < 0.05; **: *p* < 0.01) between diabetic children and healthy controls.

## Data Availability

The data presented in this study are available upon request to the corresponding author.

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
