# Peer review of "Trans Isomeric Fatty Acids in Children and Young Adults with Type 1 Diabetes Mellitus"

_nutrients, 2025, doi:10.3390/nu17111907_

Round 1
Reviewer 1 Report
Comments and Suggestions for Authors
The present study is intriguing as it investigates the levels of trans isomeric fatty acids in children and young adults with type 1 diabetes mellitus—an area that presents significant challenges for research. It is well established in the literature that insulin acts on hormone-sensitive lipoprotein lipase (LPL), which is located within adipose tissue and is responsible for hydrolyzing triglycerides stored in adipocytes. Changes in circulating insulin can influence LPL activity, and this response can vary across fat tissues. Type 1 diabetes mellitus is characterized by an absolute deficiency of insulin.
Supplementary Table S1 indicates that triglyceride levels were elevated compared to controls, although this difference was not statistically significant. Additionally, the HbA1c level was reported at 9.0%, indicating that the patients were poorly controlled, which typically correlates with elevated triglycerides and decreased HDL levels (the latter not mentioned in the table). How do the authors explain this discrepancy?
Furthermore, the authors suggest that “lower TFA values in young adults and children with diabetes may reflect a healthier diet due to the disease.” However, if the patients are poorly controlled, this may imply non-compliance with dietary recommendations or insufficient insulin dosages.
Overall, while the methodology is appropriate and the manuscript is well written, the conclusions drawn are not adequately supported by the presented data.
Author Response
First of all, we would like to thank Reviewer 1 for the helpful comments, which helped us to improve the manuscript. Please find below our point-by-point answers to the comments.
Comment 1: Supplementary Table S1 indicates that triglyceride levels were elevated compared to controls, although this difference was not statistically significant. Additionally, the HbA1c level was reported at 9.0%, indicating that the patients were poorly controlled, which typically correlates with elevated triglycerides and decreased HDL levels (the latter not mentioned in the table). How do the authors explain this discrepancy?
Answer: Thank you for this question. In these studies, we included diabetic children and young adults without comorbidities, but elevated triglycerides or suboptimal control of diabetes were no exclusion criteria. Both triglyceride and HbA1c values varied in a relative wide range, which variability is a further limitation of the study (we have added this fact to the limitations part; lines 413-7). However, there were no correlations between trans fatty acid values and HbA1c or blood sugar values, so we cannot say that poorly controlled diabetic patients might be prone to higher TFA intakes due to dietary errors. Further studies are needed to show whether diabetes control, or blood lipids may be associated with plasma and erythrocyte TFA levels in T1DM patients (lines 444-450).
Comment 2: Furthermore, the authors suggest that “lower TFA values in young adults and children with diabetes may reflect a healthier diet due to the disease.” However, if the patients are poorly controlled, this may imply non-compliance with dietary recommendations or insufficient insulin dosages.
Answer: Thank you for pointing this out, therefore we changed the conclusions from “healthier diet” to “diet lower in TFA” (lines 30 and 439).
Comment 3: Overall, while the methodology is appropriate and the manuscript is well written, the conclusions drawn are not adequately supported by the presented data.
Answer: Thank you for your helpful comments. The discussion section has been revised; now we elaborated further the potential background of the assumption that the intake of dietary TFA may be lower in patients with T1DM, based on plasma and erythrocyte membrane TFA levels (lines 285-8).
Reviewer 2 Report
Comments and Suggestions for Authors
Comments
The paper contributes valuable insights into the role of trans fatty acids in the dietary patterns of young people with type 1 diabetes. With some improvements in the clarity of data presentation, broader discussion of confounding factors, and a more thorough examination of dietary intake, this study could have a stronger impact. Consider the suggestions for improvement to enhance the paper's overall quality and contribution to the field.
Introduction
- While the introduction describes the negative effects of iTFAs on cardiovascular health, it would benefit from a stronger direct link to T1DM. For example, you could elaborate on how T1DM patients may have altered lipid metabolism and how this could interact with TFA intake. This would strengthen the rationale for why investigating TFA levels in T1DM patients is particularly important.
- The introduction could provide a clearer explanation of why the specific design (i.e., reanalyzing data from three previous case-control studies) was chosen. A brief mention of how this approach builds on previous work or allows for a deeper analysis of TFAs would help the reader understand the study’s methodology better.
- The introduction mentions that the study focuses on "young diabetic patients," but it would be helpful to define the age ranges more clearly. Is this focused only on children and young adults, or does it also include older adolescents or specific subgroups?
Methods
- The methods briefly describe the sample collection process but could provide more detail on the logistics of sample handling. For example, how were the samples stored prior to analysis? Were there any steps to minimize degradation or contamination? A clear explanation of the storage conditions (e.g., freezing, refrigeration) would enhance the reproducibility and rigor of the study.
- While the methods mention that data were re-analyzed from previous studies, there is limited explanation of how participants were selected for these studies. A more detailed description of the inclusion and exclusion criteria for both the T1DM patients and the healthy controls would improve the transparency of the study design. Were all diabetic participants receiving insulin therapy? Were there any other medical conditions that could confound the results?
- Although the study provides the number of participants in each group, it would be helpful to mention if a sample size calculation was performed prior to the original studies. This would give insight into whether the sample sizes were statistically powered to detect meaningful differences. Additionally, given that some groups (e.g., children with diabetic ketoacidosis) had smaller sample sizes, this should be addressed as a potential limitation in the study’s statistical power.
Results
- Although the results are well-presented, the section could be more focused in summarizing the most important findings. For example, instead of listing all individual TFA values in each lipid fraction, consider focusing on the most significant findings or trends across the groups. This would make the results section more concise and easier to follow.
- The results report p values but do not always explain the practical significance of these findings. For example, while some differences between the T1DM groups and controls are significant, the magnitude of the differences (effect sizes) is not discussed. It would be useful to briefly mention the clinical or biological relevance of these differences. Are the observed differences in TFA levels large enough to have potential health implications?
- This section could also mention negative or null results more explicitly. For example, if there are any lipid fractions where no significant differences were found between diabetic groups and controls, this should be mentioned to avoid the perception that the study only focused on significant results.
- The results for certain groups (e.g., diabetic children and young adults) are presented in great detail, while others (e.g., DKA group) are less thoroughly discussed. Ensure that results for all groups are equally well presented. For instance, while the DKA group is mentioned as having higher TFA levels, there is less discussion of the specific lipid fractions where these differences occurred. It would be beneficial to provide more detail on these findings.
5.Although the statistical tests used are appropriate, it is important to consider the power of the tests given the sample sizes. The smaller sample size for certain groups, such as the DKA group (n=9), could potentially lead to false negatives or reduced statistical power. A brief mention of this limitation in the results section could help contextualize the findings.
Discussion
- Although the discussion acknowledges the lack of dietary data, it does not deeply explore other potential confounding factors. For instance, factors such as the duration of diabetes, insulin therapy, or differences in physical activity could also influence TFA levels. Addressing these factors in more detail would enhance the study's depth and validity.
- The authors discuss the findings in the context of the Hungarian population but do not explore whether the results would be applicable to other populations with different dietary patterns, regulations, or genetic backgrounds. A brief mention of the external validity of the findings (e.g., comparing the results to those from other countries) would strengthen the study’s generalizability.
- The discussion primarily focuses on one hypothesis (healthier diet due to disease), but it could benefit from consideration of other possible explanations. For instance, could the differences in TFA levels reflect genetic factors, variations in gut microbiota, or differences in the degree of glycemic control among participants? Addressing alternative hypotheses would enrich the discussion.
- While the authors note the limitations of the study, they could benefit from a more explicit section on future research directions. For example, future studies could focus on dietary interventions, longitudinal monitoring of TFA levels in T1DM patients, or randomized controlled trials to directly assess the impact of reducing TFA intake on metabolic outcomes.
Conclusion
The conclusion primarily focuses on individual patient management, but it could benefit from a broader discussion on public health implications. For instance, what role might these findings play in shaping public health policies or guidelines for T1DM patients, particularly with regard to TFA consumption?
Author Response
First of all, we would like to thank Reviewer 2 for the helpful comments, which helped us to improve the manuscript. Please find below our point-by-point answers to the comments.
Introduction
- While the introduction describes the negative effects of iTFAs on cardiovascular health, it would benefit from a stronger direct link to T1DM. For example, you could elaborate on how T1DM patients may have altered lipid metabolism and how this could interact with TFA intake. This would strengthen the rationale for why investigating TFA levels in T1DM patients is particularly important.
Answer: Thank you for this insightful comment. While a large number of studies have investigated the role of TFAs in dyslipidaemia, atherosclerosis and CVD in general and in obese and/or T2DM patients, only a limited number of studies have focused on TFAs in T1DM patients. A sentence has been incorporated into the text to provide a rationale for the study (lines 72-75).
- The introduction could provide a clearer explanation of why the specific design (i.e., reanalyzing data from three previous case-control studies) was chosen. A brief mention of how this approach builds on previous work or allows for a deeper analysis of TFAs would help the reader understand the study’s methodology better.
Answer: Thank you for this helpful comment, we have modified the introduction section (lines 95-8).
- The introduction mentions that the study focuses on "young diabetic patients," but it would be helpful to define the age ranges more clearly. Is this focused only on children and young adults, or does it also include older adolescents or specific subgroups?
Answer: In these studies we included children between 1-18 years and young adults between 18-32 years old. We have added a section in the Methods part about age range of included patients. We didn’t divide them into specific subgroups because this would have resulted in too few patients in each subgroup (lines 113-130).
Methods
- The methods briefly describe the sample collection process but could provide more detail on the logistics of sample handling. For example, how were the samples stored prior to analysis? Were there any steps to minimize degradation or contamination? A clear explanation of the storage conditions (e.g., freezing, refrigeration) would enhance the reproducibility and rigor of the study.
Answer: We have added a brief description about sample handling and storage. After blood collection, the samples were transported to our laboratory within 30 minutes, where the plasma was immediately removed after centrifugation and after three washes the red blood cell mass was hemolyzed and both were stored in a -80°C freezer until further analysis (lines 126-133 and 136-9).
- While the methods mention that data were re-analyzed from previous studies, there is limited explanation of how participants were selected for these studies. A more detailed description of the inclusion and exclusion criteria for both the T1DM patients and the healthy controls would improve the transparency of the study design. Were all diabetic participants receiving insulin therapy? Were there any other medical conditions that could confound the results?
Answer: We have added a detailed description of included T1DM patients and their healthy controls in the Methods section (lines 117-126). All diabetic patients were on insulin therapy, and they had no other medical comorbidities.
- Although the study provides the number of participants in each group, it would be helpful to mention if a sample size calculation was performed prior to the original studies. This would give insight into whether the sample sizes were statistically powered to detect meaningful differences. Additionally, given that some groups (e.g., children with diabetic ketoacidosis) had smaller sample sizes, this should be addressed as a potential limitation in the study’s statistical power.
Answer: Thank you for this comment. There were no sample size calculations for these studies. We assume that 40 and 34 are conventional and probably acceptable sample sizes in studies on T1DM, whereas the number of diabetic children with DKA was much smaller. We included the relatively low sample size among the limitations of the study.
Results
- Although the results are well-presented, the section could be more focused in summarizing the most important findings. For example, instead of listing all individual TFA values in each lipid fraction, consider focusing on the most significant findings or trends across the groups. This would make the results section more concise and easier to follow.
Answer: Thank you for your helpful comment, we made a few modifications in the Results section.
- The results report p values but do not always explain the practical significance of these findings. For example, while some differences between the T1DM groups and controls are significant, the magnitude of the differences (effect sizes) is not discussed. It would be useful to briefly mention the clinical or biological relevance of these differences. Are the observed differences in TFA levels large enough to have potential health implications?
Answer: Thank you for pointing this out. We have added two sentences about the possible clinical relevance: “However, it is important to emphasize that both groups had very low levels of total TFA in all lipid fractions, so the statistically significant difference may have little physiological significance. But the present cross-sectional study design is inadequate for answering this question, longer follow-up studies would be required to address it.” (lines 288-292)
- This section could also mention negative or null results more explicitly. For example, if there are any lipid fractions where no significant differences were found between diabetic groups and controls, this should be mentioned to avoid the perception that the study only focused on significant results.
Answer: For both diabetic groups, we also wrote a sentence in the results section about non-significant findings (lines 171-3 and 191-3).
- The results for certain groups (e.g., diabetic children and young adults) are presented in great detail, while others (e.g., DKA group) are less thoroughly discussed. Ensure that results for all groups are equally well presented. For instance, while the DKA group is mentioned as having higher TFA levels, there is less discussion of the specific lipid fractions where these differences occurred. It would be beneficial to provide more detail on these findings.
Answer: Thank you for your helpful comment, we made a few modifications in this part.
Although the statistical tests used are appropriate, it is important to consider the power of the tests given the sample sizes. The smaller sample size for certain groups, such as the DKA group (n=9), could potentially lead to false negatives or reduced statistical power. A brief mention of this limitation in the results section could help contextualize the findings.
Answer: As the DKA group had a smaller number of children than the other two groups, the p-values were statistically adjusted (Bonferroni correction) to reduce the possibility of bias. This fact is addressed in the limitations section, which specifically discusses the discrepancy in the number of children in the three groups in this particular comparison (lines 419-422).
Discussion
- Although the discussion acknowledges the lack of dietary data, it does not deeply explore other potential confounding factors. For instance, factors such as the duration of diabetes, insulin therapy, or differences in physical activity could also influence TFA levels. Addressing these factors in more detail would enhance the study's depth and validity.
Answer: Thank you for this helpful comment, we have addressed these factors in the discussion section (lines 446-450).
- The authors discuss the findings in the context of the Hungarian population but do not explore whether the results would be applicable to other populations with different dietary patterns, regulations, or genetic backgrounds. A brief mention of the external validity of the findings (e.g., comparing the results to those from other countries) would strengthen the study’s generalizability.
Answer: Thank you for your question. We were able to compare the results to an Italian (Vitale et al 2013) and a Russian study (Akmurzina et al, 2013) investigating T1DM patients. As we wrote in the discussion section, our TFA values were somewhat lower than in the Italian young adults. Unfortunately enough, Akmurzina et al used μmol/l as unit, so the two set of results are not really comparable.
- The discussion primarily focuses on one hypothesis (healthier diet due to disease), but it could benefit from consideration of other possible explanations. For instance, could the differences in TFA levels reflect genetic factors, variations in gut microbiota, or differences in the degree of glycemic control among participants? Addressing alternative hypotheses would enrich the discussion.
Answer: To the best of our knowledge, diet represents the sole source of trans fatty acids in the blood; however, at this time we lack any evidence to support the hypothesis that other influencing factors may exist. Unfortunately, the literature on this topic is rather limited. In the discussion section of this manuscript, the few articles that have been found on this topic are listed. These articles discuss the duration of diabetes and the possible role of the microbiome in influencing diabetes. We also added a recent article about difference in microbiome in T1DM patients compared to healthy controls and T2DM patients (lines 349-354). It is evident that a more comprehensive set of studies is required to provide a definitive response to this question.
- While the authors note the limitations of the study, they could benefit from a more explicit section on future research directions. For example, future studies could focus on dietary interventions, longitudinal monitoring of TFA levels in T1DM patients, or randomized controlled trials to directly assess the impact of reducing TFA intake on metabolic outcomes.
Answer: We are grateful for this comment, we have added some future directions for future studies (lines 442-450).
Conclusion
The conclusion primarily focuses on individual patient management, but it could benefit from a broader discussion on public health implications. For instance, what role might these findings play in shaping public health policies or guidelines for T1DM patients, particularly with regard to TFA consumption?
Answer: We are grateful for your helpful comments. Based on our results we can corroborate the actual guideline indicating that patients with T1DM should consume the lowest possibly amount of TFAs. As a result of recent years' regulations, more and more countries are now strictly regulating the TFA content of food, so hopefully TFA intake will also decrease worldwide.
The manuscript was also reviewed by a fluent English speaker, who made changes to enhance its clarity.
Reviewer 3 Report
Comments and Suggestions for Authors
The study investigates an important and current issue: the role of trans fatty acids in the development of childhood and adolescent type 1 diabetes, with particular emphasis on blood fatty acid profiles, as well as interactions between TFAs and long-chain polyunsaturated fatty acids. The Hungarian sample provides valuable international comparison opportunities. It is important to note that the estimation of TFA intake is not based on dietary data but relies solely on plasma values.
Do the authors plan to apply a dietary questionnaire in a future study to provide a more accurate picture of the sources of TFA intake and help better understand its consumption?
The study mentions that strict TFA regulations have been in place in Hungary since 2013.
Is there any domestic research or data available on the effectiveness of this regulation in reducing the consumption of TFA-containing foods among the population?
Overall, the study is very well-structured,
but what are the key clinical takeaways, dietary recommendations, or public health actions the authors consider most important based on their findings?
Author Response
First of all, we would like to thank Reviewer 3 for the helpful comments, which helped us to improve the manuscript. Please find below our point-by-point answers to the comments.
The study investigates an important and current issue: the role of trans fatty acids in the development of childhood and adolescent type 1 diabetes, with particular emphasis on blood fatty acid profiles, as well as interactions between TFAs and long-chain polyunsaturated fatty acids. The Hungarian sample provides valuable international comparison opportunities. It is important to note that the estimation of TFA intake is not based on dietary data but relies solely on plasma values.
Comment 1: Do the authors plan to apply a dietary questionnaire in a future study to provide a more accurate picture of the sources of TFA intake and help better understand its consumption?
Answer: Thank you for this question. In our next study, we plan to assess the diet of Hungarian T1DM children using a detailed dietary questionnaire to answer the questions left open in this article.
Comment 2: The study mentions that strict TFA regulations have been in place in Hungary since 2013. Is there any domestic research or data available on the effectiveness of this regulation in reducing the consumption of TFA-containing foods among the population?
Answer: The most recent research on the content of trans fatty acids (TFA) in foods indicates that none of the investigated foods exceeded the permitted levels of TFA (https://ogyei.gov.hu/egyeb_nyilvantartasok_listak). In light of our preceding research, a decline in the total amount of TFAs in human milk has also been observed (2.06 w/w% in 2004 vs. 1.04 w/w% in 2015). However, to the best of our knowledge, no large-scale studies have been conducted in Hungary to examine the impact of TFA regulation on the plasma or erythrocyte TFA levels in children and adults.
Comment 3: Overall, the study is very well-structured, but what are the key clinical takeaways, dietary recommendations, or public health actions the authors consider most important based on their findings?
Answer: We are grateful for your helpful comments. Based on our results we can corroborate the actual guideline recommending that patients with T1DM should consume the lowest possibly amount of TFAs. As a result of recent years' regulations, more and more countries are now strictly regulating the TFA content of food, so hopefully TFA intake will also decrease worldwide.
Reviewer 4 Report
Comments and Suggestions for Authors
Thank you for sending your manuscript. Below, I outline the main concerns that led to this decision. 1.Study design and population The study appears to have significant limitations in its design, particularly with regard to the study population. The choice to focus on individuals already diagnosed with diabetes and undergoing controlled dietary interventions is problematic. The main problem is that the dietary behaviours and health outcomes of these participants are strongly influenced by the medical management they receive. This makes it difficult to isolate the impact of trans fats from other variables, such as medication and dietary control. The results would be more insightful and meaningful if the study had been conducted in a healthier, non-diabetic population to follow the potential onset of diabetes over time. A longitudinal design following individuals without diabetes, monitoring their exposure to trans fats and other dietary factors, would provide a clearer picture of how these elements may influence the development of the disease. 2.Lack of control of external factors One of the critical shortcomings of the study is the lack of consideration of external factors influencing diet and body composition, particularly among individuals with pre-existing conditions such as diabetes. These factors, such as physical activity level, medication adherence and variations in dietary control, could significantly confound the results. The current cross-sectional design does not adequately account for these variables and it is unclear how the observed results can be attributed to trans fat intake alone. 3.Relevance of the results Given the study design, the results may not be directly applicable to a larger population. Since the diabetic participants already manage their condition with controlled diets, the results do not provide indications as to how trans fats may contribute to the development of diabetes. It would be more useful to conduct studies on a population of healthy individuals and follow changes in diet, trans fat intake and metabolic health over time. 4.Statistical and methodological issues Although statistical methods are appropriate, the sample size and the use of existing cross-sectional data do not allow causal conclusions to be drawn on the relationship between trans fats and diabetes progression. The conclusions of the study might be overstated, as the study design cannot conclusively prove causality.
5. Too many self-citations
Author Response
First of all, we would like to thank Reviewer 4 for the helpful comments, which helped us to improve the manuscript. Please find below our point-by-point answers to the comments.
1.Study design and population The study appears to have significant limitations in its design, particularly with regard to the study population. The choice to focus on individuals already diagnosed with diabetes and undergoing controlled dietary interventions is problematic. The main problem is that the dietary behaviours and health outcomes of these participants are strongly influenced by the medical management they receive. This makes it difficult to isolate the impact of trans fats from other variables, such as medication and dietary control. The results would be more insightful and meaningful if the study had been conducted in a healthier, non-diabetic population to follow the potential onset of diabetes over time. A longitudinal design following individuals without diabetes, monitoring their exposure to trans fats and other dietary factors, would provide a clearer picture of how these elements may influence the development of the disease.
Answer: Thank you for this comment. Unfortunately, we couldn’t find any longitudinal studies about TFA consumption and later development of T1DM. In Finnish (Niinistö S et al, 2014; doi:10.1017/S0007114513003073; Niinistö S et al, 2015; doi: 10.1007/s00592-014-0673-0) and American (Fronczak CM et al, 2003; doi:10.2337/diacare.26.12.3237) cohort studies maternal fatty acid intake and T1DM in the offspring was investigated, but they didn’t report maternal TFA intake. Only one study stated, that maternal processed meat intake (which is a possible TFA-source) may increase the development of T1DM in the offspring, so further studies are needed to elucidate this question.
2.Lack of control of external factors One of the critical shortcomings of the study is the lack of consideration of external factors influencing diet and body composition, particularly among individuals with pre-existing conditions such as diabetes. These factors, such as physical activity level, medication adherence and variations in dietary control, could significantly confound the results. The current cross-sectional design does not adequately account for these variables and it is unclear how the observed results can be attributed to trans fat intake alone.
Answer: We are grateful for this commentary. Within the Discussion section at the conclusions of the article, it is now stated that further studies are required to ascertain whether confounders have any role to play in the levels of TFAs observed in patients diagnosed with T1DM (line 442-450).
3.Relevance of the results Given the study design, the results may not be directly applicable to a larger population. Since the diabetic participants already manage their condition with controlled diets, the results do not provide indications as to how trans fats may contribute to the development of diabetes. It would be more useful to conduct studies on a population of healthy individuals and follow changes in diet, trans fat intake and metabolic health over time.
Answer: Thank you for this comment. There are currently few studies available that investigate the role of trans fatty acids in the development of T1DM. These are mainly cross-sectional and not longitudinal studies. Previous studies have mainly focused on the development of insulin resistance, but a recent in vitro cell study (Szustak et al, 2024) raised the possibility of direct toxicity of trans-palmitoleic acid on pancreatic beta cells. This article is cited in the discussion section (line 356-362)
4.Statistical and methodological issues Although statistical methods are appropriate, the sample size and the use of existing cross-sectional data do not allow causal conclusions to be drawn on the relationship between trans fats and diabetes progression. The conclusions of the study might be overstated, as the study design cannot conclusively prove causality.
Answer: We would like to express our gratitude for the comments that have been provided. The conclusions section in the abstract and at the end of the article has been modified accordingly.
- Too many self-citations
Answer: We have deleted the unnecessary self-citations.
Round 2
Reviewer 1 Report
Comments and Suggestions for Authors
no other comments
Author Response
We would like to thank you again for the constructive comments that let us improve our manuscript.
Reviewer 3 Report
Comments and Suggestions for Authors
Thank you. I accept the revised version of the manuscript.
Reviewer 4 Report
Comments and Suggestions for Authors
I would like to thank the authors for their responses. However, I must respectfully point out that my main concerns have not been adequately addressed and no substantial changes appear to have been made to the manuscript based on the comments provided.
Study design and population
The manuscript still lacks a critical discussion of the limitations of the study design. The influence of medical management (dietary recommendations, insulin therapy, etc.) on the eating behavior of patients with T1DM has not been considered. Unless this is adequately addressed, the findings suggesting that diabetic subjects have lower TFA intake cannot be interpreted as reflecting autonomous food choices. This issue continues to undermine the interpretability of the results.
Confounding Factors
The manuscript does not include any serious consideration of key external confounding factors, such as physical activity, medication adherence, or other lifestyle factors. The simple statement that “further studies are needed” is not sufficient. These variables are crucial and should at least be acknowledged explicitly in the discussion of limitations, not just mentioned briefly.
Relevance of the findings
The conclusions remain exaggerated. While acknowledging the cross-sectional design, the authors continue to speculate on causality and make public health suggestions (e.g., “there is no urgency to recommend further reduction in TFA intake”). Given the lack of data on dietary TFA intake and the observational nature of the study, these statements are unwarranted and should be eliminated or reworded more cautiously.
Methodological issues and interpretation
Although statistical methods may be appropriate, limitations related to sample size and study design are not adequately reflected in the abstract or conclusions. More cautious language is warranted throughout the text.
Self-quotations
Despite the authors' statement, the manuscript still contains a large number of self-citations. Some appear unnecessary or only marginally relevant. This aspect should be reviewed more critically.
Author Response
First of all, we would like to thank Reviewer 4 for the helpful comments, which helped us to further improve the manuscript. Please find below our point-by-point answers to the comments.
Study design and population
The manuscript still lacks a critical discussion of the limitations of the study design. The influence of medical management (dietary recommendations, insulin therapy, etc.) on the eating behavior of patients with T1DM has not been considered. Unless this is adequately addressed, the findings suggesting that diabetic subjects have lower TFA intake cannot be interpreted as reflecting autonomous food choices. This issue continues to undermine the interpretability of the results.
Answer: Thank you for this insightful comment. TFAs come from the diet, but we agree with Reviewer 4 that without a detailed FFQ we cannot interpret our results as healthier food choices of T1DM patients. We couldn’t find any literature about the possible role of type of insulin treatment on fatty acid supply, although a difference between insulin pump and conventional insulin therapy may be present. We addressed these limitations in the Discussion part (lines 336-346).
Confounding Factors
The manuscript does not include any serious consideration of key external confounding factors, such as physical activity, medication adherence, or other lifestyle factors. The simple statement that “further studies are needed” is not sufficient. These variables are crucial and should at least be acknowledged explicitly in the discussion of limitations, not just mentioned briefly.
Answer: Thank you for highlighting this important point. We acknowledge that several confounding factors may influence fatty acid composition, including TFAs. However, the number of studies directly addressing this topic remains limited. While both acute and regular physical activity are known to affect fatty acid profiles, we were unable to identify any studies that specifically examined these effects in patients with T1DM. Similarly, we found no literature addressing the influence of medical adherence or the type of insulin regimen on fatty acid composition. In response to your comment, we have now addressed these potential confounders in the Discussion section (lines 336–346).
Relevance of the findings
The conclusions remain exaggerated. While acknowledging the cross-sectional design, the authors continue to speculate on causality and make public health suggestions (e.g., “there is no urgency to recommend further reduction in TFA intake”). Given the lack of data on dietary TFA intake and the observational nature of the study, these statements are unwarranted and should be eliminated or reworded more cautiously.
Answer: We deleted the suggested part and revised in a more cautious way (lines 443-445).
Methodological issues and interpretation
Although statistical methods may be appropriate, limitations related to sample size and study design are not adequately reflected in the abstract or conclusions. More cautious language is warranted throughout the text.
Answer: Based on the suggestions we have revised the conclusion part of the manuscript (lines 29-31 and 436-445).
Self-quotations
Despite the authors' statement, the manuscript still contains a large number of self-citations. Some appear unnecessary or only marginally relevant. This aspect should be reviewed more critically.
Answer: Out of 40 references 5 citations are self-citations. As indicated in the body of the text, three of these articles (Nr 25, 26 and 27) are prior communications that form the basis of the present manuscript and must be cited therein.
As discussed in detail in our previous review (Hatem et al, Front Nutr 2024, doi: 10.3389/fnut.2024.1379772), few studies to date have investigated the relationship between cis and trans fatty acids in human milk (Tinoco et al, 2008, n = 37); Krešic et al, 2013, n = 83; Bousset-Alferes et al, 2022, n = 33). Among these, our own article included here (Nr 19) is based on the largest sample size (n = 769). The study was derived from data of a birth cohort investigation, which is why we consider it the most relevant reference in the literature.
Our last article (Nr 32) is found in the Discussion section, where we examine the potential relationship between the gut microbiome and fecal trans fatty acid content. While the link between the microbiome and short-chain fatty acids is well established, to the best of our knowledge, our experimental study is among the first in the literature to demonstrate a direct link between the gut microbiome and the detailed fecal trans fatty acid (and other fatty acid) levels. Our findings complement the hypothesis proposed in the preceding article, underscoring the necessity for further research in this area. Recent studies have raised the possibility that dietary fatty acids, including trans fatty acids, may influence the development of certain diseases (such as inflammatory bowel disease) or cognitive functions via the gut-brain axis. This suggests that dietary fatty acids may exert its effects not only directly, through absorption from the gut and incorporated into cells, but also indirectly, through modulation of the gut microbiome, an emerging field of growing scientific interest.